# Volatile Compounds in Fruit Peels as Novel Biomarkers for the Identification of Four Citrus Species

**DOI:** 10.3390/molecules24244550

**Published:** 2019-12-12

**Authors:** Haipeng Zhang, Huan Wen, Jiajing Chen, Zhaoxin Peng, Meiyan Shi, Mengjun Chen, Ziyu Yuan, Yuan Liu, Hongyan Zhang, Juan Xu

**Affiliations:** Key Laboratory of Horticultural Plant Biology (Ministry of Education), College of Horticulture and Forestry, Huazhong Agricultural University, Wuhan 430070, China; haipengzhang@webmail.hzau.edu.cn (H.Z.); ndwenwen@163.com (H.W.); jiajingchen@webmail.hzau.edu.cn (J.C.); pengzhaoxin@webmail.hzau.edu.cn (Z.P.); shimeiyan@webmail.hzau.edu.cn (M.S.); c18186100353@163.com (M.C.); yuanziyu@webmail.hzau.edu.cn (Z.Y.); luiyuer@webmail.hzau.edu.cn (Y.L.); zhanghy@mail.hzau.edu.cn (H.Z.)

**Keywords:** citrus, volatiles, biomarkers

## Abstract

The aroma quality of citrus fruit is determined by volatile compounds, which bring about different notes to allow discrimination among different citrus species. However, the volatiles with various aromatic traits specific to different citrus species have not been identified. In this study, volatile profiles in the fruit peels of four citrus species collected from our previous studies were subjected to various analyses to mine volatile biomarkers. Principal component analysis results indicated that different citrus species could almost completely be separated. Thirty volatiles were identified as potential biomarkers in discriminating loose-skin mandarin, sweet orange, pomelo, and lemon, while 17 were identified as effective biomarkers in discriminating clementine mandarins from the other loose-skin mandarins and sweet oranges. Finally, 30 citrus germplasms were used to verify the classification based on β-elemene, valencene, nootkatone, and limettin as biomarkers. The accuracy values were 90.0%, 96.7%, 96.7%, and 100%, respectively. This research may provide a novel and effective alternative approach to identifying citrus genetic resources.

## 1. Introduction

*Citrus* is one of the most important fruit tree genera in the world; *Citrus* acreage and production are top-ranked globally [1]. Citrus fruit are well accepted by consumers because of their nutritional and health-promoting properties. In citrus fruit, bioactive compounds, including vitamin C, soluble sugars, organic acids, amino acids, flavonoids, carotenoids, and volatile compounds, contribute to the global fruit quality while also being beneficial to human health [1,2,3,4,5,6,7,8]. Other than their contribution to the odor notes, volatile compounds play important roles in the interactions between plants and their environment [9] via attracting pollinators and defending against pathogens and herbivores [10,11,12].

Citrus fruit peels are rich in volatile compounds, mainly including terpenoids, aldehydes, alcohols, acids, and esters [8]. Terpenoids are the most abundant volatiles, accounting for more than 90% in most citrus germplasms [8]. Both the composition and contents of volatile compounds are important in affecting the fruit odor and sensory properties. For instance, the contents of 16 volatile compounds were significantly changed in the fruit of Niurouhong (*Citrus reticulata* Blanco) in comparison with its wild type, Zhuhongju, resulting in significant changes in the aroma traits [13]. The contents of *cis*- and *trans*-linalool oxides in Huanong Red pomelo fruit increased after being pollinated with *Citrus mangshanensis*, which consequently obtained the characteristic aroma of its pollen parent [14]. By using partial least squares discriminant analysis (PLS-DA), 15 compounds were selected as metabolite markers to distinguish between Chenpi (*C. reticulata* Blanco) and Guangchenpi (*C. reticulata* ‘Chachi’) fruit peel [15].

As one of the original centers of *Citrus*, China has many citrus germplasm resources, mainly including sweet orange (*C. sinensis*), loose-skin mandarin (*C. reticulata*), lemon (*C. limon*), and pomelo (*C. grandis*) [1,16]. Furthermore, some hybrid citrus germplasms, such as clementine mandarin (*C. clementina* Hort. ex Tanaka), which may be a hybrid of loose-skin mandarin and sweet orange [17,18], are also widely cultivated. It is known that the fruit of each citrus species has a recognizable aroma trait. For example, the aromas of fruit are quite similar in different sweet orange germplasms but different from loose-skin mandarin, pomelo, and lemon accessions. The aroma of clementine mandarin is different from both parents; however, to our knowledge, the specific volatile compounds accounting for the differences have not been identified.

Previous reports have focused on the identification of key characteristic aroma compounds that contribute to the specific flavor [19,20,21,22,23]. In these reports, valencene was taken as a characteristic aromatic compound in sweet orange fruit [23], *cis*- and *trans*-linalool oxides and β-myrcene were the main characteristic aromatic compounds in *C. mangshanensis* fruit [21], β-pinene, γ-terpinene, linalyl acetate, and linalool contributed to the characteristic aroma of lemon fruit [22], β-citronellal, nerol acetate, and geranyl acetate were mainly responsible for the characteristic aroma of lime fruit [20,22], and linalool and its derivatives can largely reconstruct the sweet and fragrant aroma of the Miyamoto satsuma mandarin (*C. reticulata*) [19]. All of these characteristic volatiles can serve as biomarkers to distinguish the specific citrus germplasm. However, to our knowledge, these studies only focused on limited citrus germplasms, and the characteristic aromatic compounds may not be species specific.

Hierarchical clustering analysis (HCA), principal component analysis (PCA), and PLS-DA have been effectively used to analyze the relationship between different citrus species based on metabolomics data [8,24,25]. PLS-DA has been used to determine the differentially accumulated compounds in different species, where the compounds with a high variable importance in projection (VIP) value were selected as the potential specific biomarkers [15].

In the study, large-scaled volatile compound data from 66 citrus germplasms representing four citrus species were used to mine the potential species-specific volatile biomarkers. PCA and PLS-DA were applied in choosing the important volatiles that contribute to the specific volatile profile of different citrus species, and a total of 30 compounds were selected as potential biomarkers. Thirty citrus germplasms were used for classification, and a high level of accuracy was found, based on four volatile compounds with high VIP values.

## 2. Results

### 2.1. Volatile Compounds in Fruit Peels

In total, 89 volatile compounds (see Appendix A) from four citrus species were used for analysis. The Venn plot indicated that all of them were detected in at least two citrus species, and 59 volatile compounds were commonly detected in four species (Figure 1A), indicating that the volatile compounds identified were similar in different citrus species.

The PCA results showed that the first principal component explained 29% of the variance— loose-skin mandarin (LSM) and lemon (Lem) were clearly separated from pomelo (P) and sweet orange (SW) on the PC1 axis. The second principal component explained 16% of the variance—SW was clearly separated from P, while LSM was separated from Lem on the PC2 axis (Figure 1B). Although the first two principal components explained only approximately 45% of the variance, four citrus species (LSM, SW, P, and Lem) used in the study were clearly distinguished from each other. The results indicated that different citrus species were indeed species specific with regard to their volatile profiles.

### 2.2. Accumulation Pattern of Volatile Compounds is Citrus Species Dependent

To identify the potential volatile biomarkers that discriminate different citrus species, the volatile profiles of LSM, SW, P, and Lem were applied. The results indicated that almost all of them were grouped into a citrus species and separated from the other citrus species by a PLS-DA loading plot (Figure 2). In total, 30 potential volatile biomarkers were found that might be used to distinguish different citrus species (Table 1), and 17 potential volatile biomarkers contributed to the difference between clementine mandarin and its parents (LSM and SW) (see Appendix A).

#### 2.2.1. Discrimination of LSM from the Other Three Citrus Species

To discriminate LSM from the other citrus species and find the potential markers responsible for such classification, PLS-DA was employed. The PLS-DA loading plot showed that the volatile profiles of LSM and the other citrus species could be separated (Figure 2A), indicating that the volatile compositions of LSM and the other citrus species were quite different. The volatile markers were selected based on the VIP value (Table 1).

Consequently, 10 volatile compounds with a VIP value greater than 1.5 were selected as biomarker compounds that were responsible for the discrimination of LSM from the other citrus species. As shown in Figure 3A, the contents of β-elemene, germacrene B, 3-hexenal, γ-elemene, α-caryophyllene, δ-elemene, and γ-terpinene in LSM were significantly higher than those in the other citrus species, while the levels of valencene, (*Z*)-β-farnesene, and caryophyllene oxide in LSM were significantly lower than those in the other citrus species. These compounds might serve as potential biomarkers for the discrimination of LSM from the other citrus species.

#### 2.2.2. Discrimination of Sweet Orange from the Other Three Citrus Species

To find the potential biomarkers that contribute to the differences between SW and the other citrus species, the volatile profiles of 24 sweet oranges and the other 42 citrus germplasms were analyzed, and eight volatile compounds were selected as candidate biomarkers by PLS-DA (Figure 2B). The valencene and caryophyllene oxide were the most important compounds (VIP value > 2) (Table 1). A comparison between the contents of the biomarkers in SW and the other citrus species found that the levels of valencene, caryophyllene oxide, α-phellandrene, and *trans*-limonene oxide were high in SW, while the levels of γ-terpinene, α-thujene, germacrene D, and α-terpinene were low (Figure 3B).

#### 2.2.3. Discrimination of Pomelo from the Other Three Citrus Species

To investigate the difference between pomelo and the other citrus species, nine biomarker compounds were selected by PLS-DA (Figure 2C). Meanwhile, the significant differences were estimated in pomelo and the other citrus species, and six volatile compounds were identified by combining the VIP scores and P-values. Nootkatone and 3-hexenal were the most important compounds (VIP value > 2) (Table 1). Among the six biomarker compounds, the content of nootkatone in pomelos was significantly higher than that in the other citrus species, while the levels of 3-hexenal, *trans*-limonene oxide, β-cubebene, elemol, and octyl ester were low in pomelos (Figure 3C).

#### 2.2.4. Discrimination of Lemon from the Other Three Citrus Species

Eleven volatile compounds served as biomarkers to distinguish five lemons from the other 61 citrus germplasms (Figure 2D). Limettin, citronellal, *trans*-α-bergamotene, and β-bisabolene were the most important compounds (VIP value > 2) (Table 1). The contents of 11 biomarker compounds (limettin, citronellal, *trans*-α-bergamotene, β-bisabolene, *cis*-α-bergamotene, β-pinene, caryophyllene, camphene, α-cubebene, α-terpineol, and (*Z*)-β-farnesene) in lemons were significantly higher than those in the other citrus species (Figure 3D).

#### 2.2.5. Discrimination of Clementine Mandarin from LSM and SW

Clementine mandarin is a hybrid cultivar from LSM and SW [16], with a specific aroma different from LSM and SW. To identify the specific compounds that contribute to the difference, 16 clementine mandarins, 29 LSMs, and 24 SWs were used, and 16 marker compounds were selected by PLS-DA (Figure 4; see Appendix A). Dodecanal, decanal, (*Z*)-β-farnesene, α-sinensal, ylangene, α-muurolene, and α-terpineol acetate were the important compounds (VIP value > 2). It was found that the levels of all of these 16 markers were high in clementine mandarins (see Appendix A).

### 2.3. Four Biomarkers for the Identification of Four Citrus Species

Overall, β-elemene, valencene, nootkatone, and limettin were the most important compounds, and their contents were significantly different in LSM, SW, P, and Lem (Figure 5). To further verify the accuracy of the four biomarkers in the identification of four citrus species, 30 citrus germplasms collected in 2017 were used in the study. β-elemene was used as a marker to classify the 30 citrus germplasms into two groups—LSM or not LSM. The results showed that 27 citrus germplasms were correct, while the other three (Suhong tangerine, Hamlin sweet orange, and Kesai lime) were incorrect. The accuracy values were 90.0%, 96.7%, 96.7%, and 100%, based on β-elemene, valencene, nootkatone, and limettin as markers, respectively (Table 2).

### 2.4. Discrimination of Wild and Cultivar Germplasms

Twenty compounds were selected as biomarkers that contributed to the discrimination of wild and cultivar germplasms by PLS-DA. Germacrene B, γ-elemene, and *trans*-nerolidol were the important compounds with VIP values >2 (see Appendix A). The contents of 19 marker compounds were high, while only one had a low level in wild germplasms (see Appendix A).

## 3. Discussion

### 3.1. Biomarkers for Discriminating between Different Citrus Germplasms

Lots of germplasms in *Citrus*, including mainly LSM, sweet orange, pomelo, lemon, and various hybrid germplasms, were used in the study [1]. For example, the fruit shape and size of clementine mandarin were similar to LSM but had a different aroma. The contents of 14 biomarker compounds in clementine mandarin were significantly higher than those in LSM and sweet orange (see Appendix A). The results indicated that these 14 compounds may contribute to the specific aroma of clementine mandarins and can also be used as biomarkers to distinguish clementine mandarin from LSM and sweet orange. Due to more and more hybrid germplasms having been released from citrus breeding programs, it is hard to distinguish them from each other just based on fruit shape and size. The volatile biomarkers may thus be a good method for discrimination between them. In total, 30 volatile biomarkers were found to distinguish between different citrus species (Table 1).

Furthermore, four compounds with the highest VIP values were selected as biomarkers to discriminate between 30 citrus germplasms. The accuracy of the identification results was very high, with only three, one, one, and zero being incorrect, based on β-elemene, valencene, nootkatone, and limettin, respectively (Table 2). Therefore, with the use of volatile compounds as biomarkers for the identification of citrus germplasms shown to be reliable and with the method being cheaper, simpler, faster, and easier, using volatile compounds might serve as an alternative or primary method to the molecular methods (simple sequence repeats and single-nucleotide polymorphisms).

### 3.2. Biomarkers May Be Responsible for the Citrus Species-Specific Odor Notes

There were abundant volatile compounds in citrus fruit. Lots of reports have mainly focused on the determination of volatile compounds and the comparison of the number and content of them in a few citrus germplasms [8,13,21,24,25,26,27]. As different citrus species may have a unique aroma, the profiles of volatiles have been used to study citrus chemotaxonomy [8,24,25]. In addition, some researchers have identified the characteristic aroma compounds in citrus germplasms with specific aroma traits, such as *C. mangshanensis*, sweet orange, lemon, and lime [19,20,21,22,23].

It has been reported that valencene is a characteristic aroma compound in sweet orange [23,28]. In the study, a high VIP value (2.54) of valencene was found by PLS-DA (Table 1, Figure 3B). Valencene was mainly accumulated in sweet orange, indicating that valencene may contribute to the characteristic aroma of sweet orange. In addition, the levels of some biomarker compounds were high or low in SW fruit (Table 1, Figure 3B), suggesting that these compounds may also contribute to its specific aroma. Furthermore, ten, eleven, and seven biomarker compounds were selected in LSM, lemon, and pomelo, respectively (Table 1). The different accumulation of these compounds in different citrus species may be responsible for the species-specific aroma.

Although some candidate compounds that might contribute to the specific aroma were selected by PLS-DA (Table 1), the contribution of each compound was still unclear. To further determine the effects of these compounds, gas chromatography-olfactory (GC-O), aroma extraction dilution analysis (AEDA), and odor activity analysis were required [21]. The specific aroma of citrus fruit does not result from one specific volatile. For example, the specific balsamic and floral odor of *C. mangshanensis* fruit results from *d*-limonene as a background aroma, *trans*- and *cis*-linalool oxides, and β-myrcene [21].

### 3.3. Protection and Utilization of Wild Citrus Germplasms

In the citrus breeding process, the traits of fruit yield, maturity, and color are more likely to attract the attention of breeders. In flavor, the contents of sugar and acids are the most important, with the aroma trait having often been ignored in citrus breeding history. The levels of 19 of the 20 compounds in cultivars were lower than in wild germplasms. Similarly, the characteristic aroma compounds of *C. mangshanensis* (*trans*- and *cis*-linalool oxides) were detected in some wild citrus germplasms but not in cultivars [21]. Some of the compounds with decreased levels may contribute to the aroma odor in citrus fruit, such as linalool, *trans*-nerolidol, citronellal, and γ-terpinene. It was found that 18 of the 19 compounds were terpene compounds, and many terpene compounds play important roles in interactions with the environment, such as attracting pollinators and defending against pathogens and herbivores [10,12,29]. Therefore, the decreased levels of these volatile compounds might result from changes in the living environment. The wild citrus germplasms with good aroma traits may not only be used in citrus breeding but also provide raw materials for the extraction of essential oil.

## 4. Materials and Methods

### 4.1. Materials

The raw data of volatile compounds in the citrus peels were downloaded from our previous study [8]. At least five representative germplasms were selected from each citrus species, and as a result, volatile data from 29 loose-skin mandarins (LSMs), 24 sweet oranges (SWs), eight pomelos (Ps) and five lemons (Lems) (see Appendix A) were used in the study. The volatile compounds that were detected in at least five germplasms were selected. Additionally, 16 clementine mandarins were used to analyze the difference between LSM and SW.

The fruits of 30 citrus germplasms were collected from the National Citrus Breeding Center (Wuhan, Hubei, China) and the Citrus Research Institute, Chinese Academy of Agricultural Sciences (Beibei, Chongqing, China), including 10 LSMs, nine sweet oranges, six pomelos, and five lemons. The fruit peels were separated and placed in liquid nitrogen and then stored at −80 °C.

### 4.2. Extraction and Determination of Volatiles

According to the method of Zhang et al. [8], the volatile compounds were extracted by methyl tert-butyl ether (MTBE, HPLC grade) from 1 g of citrus peels. The TRACE GC Ultra GC coupled with a DSQ II mass spectrometer (Thermo Fisher Scientific, Waltham, MA, USA) with a TRACE TR-5 MS column (30 m × 0.25 mm × 0.25 μm; Thermo Scientific, Bellefonte, PA, USA) were used to obtain the profiles of the volatile compounds.

### 4.3. Data Analysis

A Venn plot was drawn using VENNY 2.1 [30]. Principal component analysis (PCA) by image GP (http://www.ehbio.com/ImageGP/index.php/Home/Index/index.html) was used to determine the volatile profiles of 66 citrus germplasms.

PLS-DA analysis was conducted using the R package mixOmics [31] and SIMCA-P software (Umetrics AB, Umea, Sweden). For example, to understand the difference of volatile compounds between LSM and the other citrus species, 29 LSMs and 37 other citrus germplasms were used to calculate the VIP value for each volatile compound by the PLS-DA method. A compound was selected when its VIP value was greater than 1.5. The specific compounds with high VIP values were used to distinguish between different citrus germplasms, and the difference was further verified using the volatile profiles with the R package ggplot2 and reshape2. R packages (nortest, stats, and pgirmess) were used for the ANOVA (*p* < 0.05).

To verify the accuracy of four biomarkers in discriminating between citrus species, a citrus germplasm was classified as LSM if its content of β-elemene was higher than 16 ng/g and as SW, P, or Lem when valencene, nootkatone, or limettin was detected.

## 5. Conclusions

The volatiles in the peels of 66 citrus germplasms from four citrus species were used for biomarker mining, and 30 potential biomarkers with different accumulation patterns in different citrus species were chosen using PLS-DA. The β-elemene, valencene, nootkatone, and limettin had the highest VIP values and were chosen as biomarkers for the identification of citrus species. An accuracy of 90.0%, 96.7%, 96.7%, and 100% in loose-skin mandarin, sweet orange, pomelo, and lemon was obtained, respectively. These biomarker compounds may be responsible for the specific aroma in different citrus species. This method is a novel and effective alternative approach to identifying citrus genetic resources with biomarkers.

## Figures and Tables

**Figure 1 molecules-24-04550-f001:**
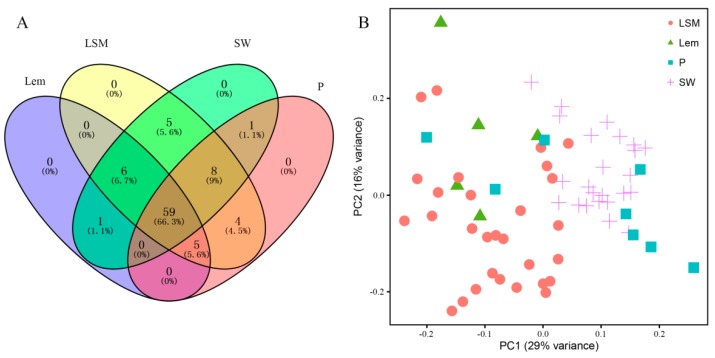
Volatile compounds in four citrus species. (**A**) Venn plot of the volatile compounds in four citrus species and (**B**) principal component analysis (PCA) results of the volatile profiles in four citrus species. SW: sweet orange; P: pomelo; LSM: loose-skin mandarin; Lem: lemon.

**Figure 2 molecules-24-04550-f002:**
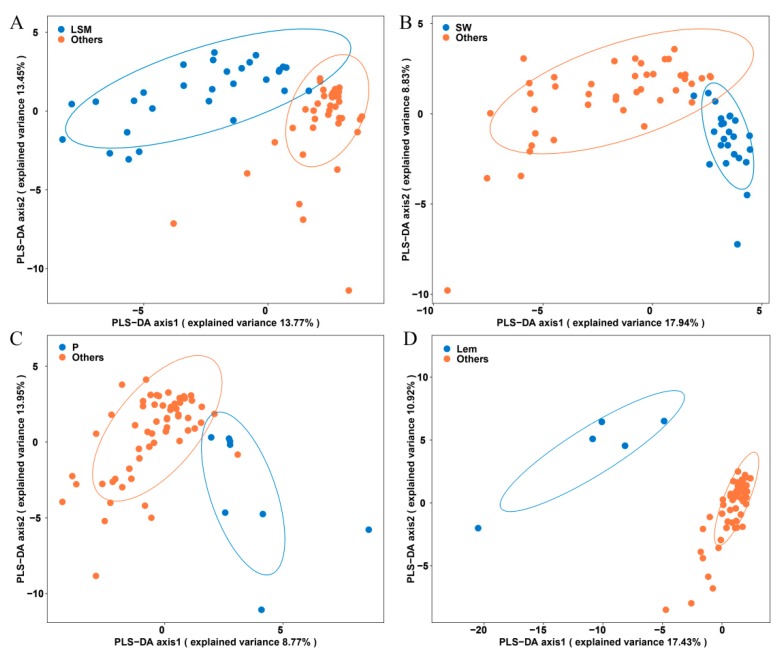
Partial least squares discriminant analysis (PLS-DA) score plots of four citrus species. (**A**) The results of the PLS-DA clearly distinguished the LSM and the other three citrus species using volatile profiles. (**B**–**D**) The results of the PLS-DA clearly distinguished the SW/P/Lem and the other three citrus species using volatile profiles, respectively. LSM: loose-skin mandarin; SW: sweet orange; P: pomelo; Lem: lemon; O: other citrus species.

**Figure 3 molecules-24-04550-f003:**
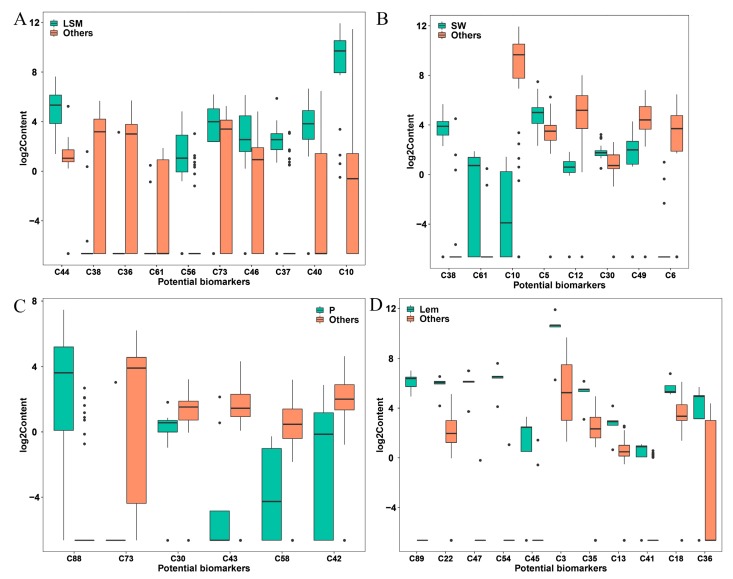
Boxplots showing the contents of biomarkers in LSM (**A**), SW (**B**), P (**C**), and Lem (**D**) and the other citrus germplasms. The biomarkers are listed in Table 1. LSM: loose-skin mandarin; SW: sweet orange; P: pomelo; Lem: lemon.

**Figure 4 molecules-24-04550-f004:**
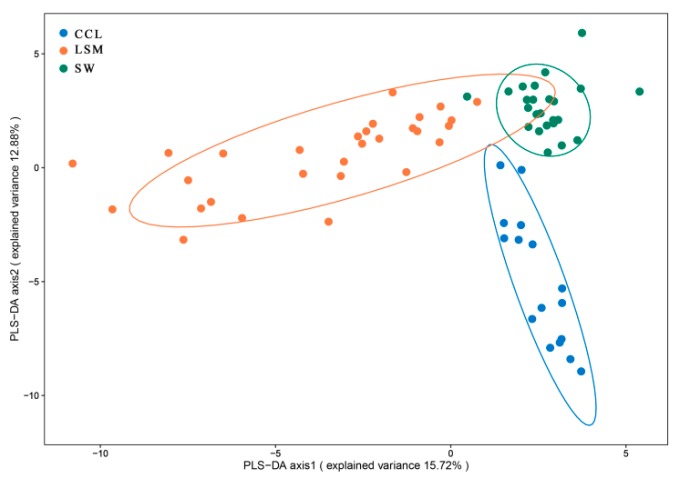
PLS-DA score plots of clementine mandarin, LSM, and SW. LSM: loose-skin mandarin; SW: sweet orange; CCL: clementine mandarin.

**Figure 5 molecules-24-04550-f005:**
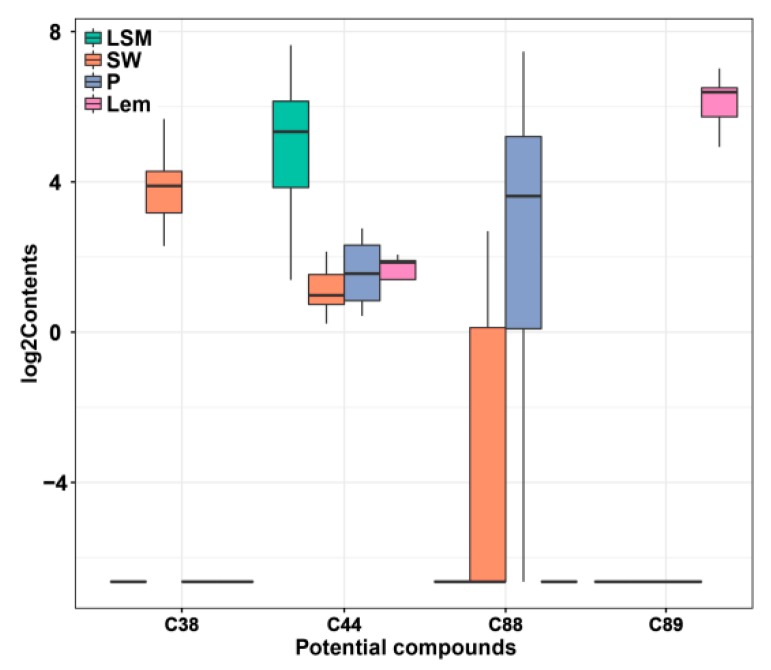
Boxplot showing the contents of four high variable importance in projection (VIP) value biomarkers in LSM, SW, P, and Lem. LSM: loose-skin mandarin; SW: sweet orange; P: pomelo; Lem: lemon. C38: valencene; C44: β-elemene; C88: nootkatone; C89: limettin.

**Table 1 molecules-24-04550-t001:** Potential biomarkers selected in four citrus species.

ID	Compounds	VIP	P-Value	ID	Compounds	VIP	*P*-Value
**Loose-skin mandarin (LSM)**				**Lemon (Lem)**			
C44	β-elemene	1.98	3.85 × 10^−^^10^	C89	limettin	2.51	8.00 × 10^−16^
C38	valencene	1.75	4.67 × 10^−6^	C22	citronellal	2.40	2.91 × 10^−4^
C36	(*Z*)-β-farnesene	1.63	7.15 × 10^−6^	C47	*trans*-α-bergamotene	2.30	1.26 × 10^−13^
C61	caryophyllene oxide	1.63	2.76 × 10^−4^	C54	β-bisabolene	2.22	1.26 × 10^−13^
C56	germacrene B	1.60	5.95 × 10^−7^	C45	*cis*-α-bergamotene	1.96	5.52 × 10^−9^
C73	3-hexenal	1.56	3.69 × 10^−2^	C3	β-pinene	1.95	1.56 × 10^−3^
C46	γ-elemene	1.54	9.47 × 10^−5^	C35	caryophyllene	1.89	9.80 × 10^−4^
C37	α-caryophyllene	1.51	7.68 × 10^−10^	C13	camphene	1.81	2.00 × 10^−3^
C40	δ-elemene	1.50	1.86 × 10^−7^	C41	α-cubebene	1.71	1.15 × 10^−4^
C10	γ-terpinene	1.50	8.75 × 10^−4^	C18	α-terpineol	1.57	1.03 × 10^−3^
**Sweet orange (SW)**				C36	(*Z*)-β-farnesene	1.56	4.24 × 10^−3^
C38	valencene	2.54	6.19 × 10^−9^	**Pomelo (P)**			
C61	caryophyllene oxide	2.44	2.90 × 10^−11^	C88	nootkanone	3.07	1.13 × 10^−5^
C10	γ-terpinene	1.61	2.16 × 10^−8^	C73	3-hexenal	2.01	1.51 × 10^−3^
C5	α-phellandrene	1.56	5.68 × 10^−5^	C30	*trans*-limonene oxide	1.87	2.70 × 10^−3^
C12	α-thujene	1.53	2.61 × 10^−8^	C43	β-cubebene	1.77	2.07 × 10^−3^
C30	*trans*-limonene oxide	1.53	9.49 × 10^−4^	C58	elemol	1.77	5.16 × 10^−4^
C49	germacrene D	1.52	1.38 × 10^−7^	C42	copaene	1.61	4.16 × 10^−3^
C6	α-terpinene	1.50	2.05 × 10^−7^				

**Table 2 molecules-24-04550-t002:** Four biomarkers for the identification of four citrus species.

Citrus Germplasms	Actual	β-Elemene ^a^	Valencene ^b^	Nootkanone ^c^	Limettin ^d^
Fu tangerine	LSM	Yes (Y)	N	N	N
Huapi tangerine	LSM	Y	N	N	N
Zhuhong tangerine	LSM	Y	N	N	N
Nian tangerine	LSM	Y	N	N	N
Shatang tangerine	LSM	Y	N	N	N
Suhong tangerine	LSM	No (N)	N	N	N
Nanfengmi tangerine	LSM	Y	N	N	N
Huanongbendizao tangerine	LSM	Y	N	N	N
Tu tangerine	LSM	Y	N	N	N
Red tangerine	LSM	Y	N	N	N
Anliu sweet orange	SW	N	Y	N	N
Valencia sweet orange	SW	N	Y	N	N
Hamlin sweet orange	SW	Y	Y	N	N
Hong anliu sweet orange	SW	N	Y	N	N
Cara cara navel orange	SW	N	Y	N	N
Jincheng sweet orange	SW	N	Y	N	N
Washington navel orange	SW	N	Y	N	N
Seika navel orange	SW	N	Y	N	N
Newhall navel orange	SW	N	Y	N	N
Feicui pomelo	P	N	N	Y	N
Kaopan pomelo	P	N	N	Y	N
Huanong red pomelo	P	N	N	Y	N
Liangping pomelo	P	N	N	Y	N
Wanbai pomelo	P	N	N	Y	N
Low-acid pomelo	P	N	N	Y	N
Finger citron	Lem	N	N	N	Y
Tahiti lime	Lem	N	N	Y	Y
Kesai lime	Lem	Y	Y	N	Y
Verna lemon	Lem	N	N	N	Y
Eureka lemon	Lem	N	N	N	Y
Accuracy		90.0%	96.7%	96.7%	100.00%

^a^ The citrus germplasm was classified as loose-skin mandarin (LSM) or not, using β-elemene as a marker; ^b^ the citrus germplasm was classified as sweet orange (SW) or not, with valencene as a marker; ^c^ the citrus germplasm was classified as pomelo (P) or not, using nootkanone as a marker; and ^d^ the citrus germplasm was classified as lemon (Lem) or not, using limettin as a marker.

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
