# Peer review of "Volatile Compounds in Fruit Peels as Novel Biomarkers for the Identification of Four Citrus Species"

_molecules, 2019, doi:10.3390/molecules24244550_

Round 1

Reviewer 1 Report

The authors raise a important research topic.

Although the results are difficult to evaluate, because the attached file ‘Supplementary Materials’  is not editable (cannot be opened).

The authors must also present the results regarding the validation of determinations using HPLC and GC methods.

Authors must complete the publication with a chapter ‘Conclusions’.

Author Response

Point1: Although the results are difficult to evaluate, because the attached file ‘Supplementary Materials’ is not editable (cannot be opened).

Response: Thanks for your suggestions. The supplementary materials were uploaded as .doc type.

Point2: The authors must also present the results regarding the validation of determinations using HPLC and GC methods.

Response: The HPLC is mainly used to analyze metabolites with high boiling point, macro-molecule and strong polarity compounds. The GC is an applicative analysis instrument to detect compounds with lower boiling points. For compounds identification, both HPLC and GC are based on authentic standards, whilst their detector sensitivity is much lower than GC-MS. Most of volatile compounds at lower levels in citrus fruit are hardly to obtain their corresponding authentic standards. However, the NIST Mass Spectral Library embedded in GC-MS analytical software facilities the identification of various trace amount metabolites. Thus, it is not an appropriate method to measure the volatile compounds by using HPLC or GC.

Point3: Authors must complete the publication with a chapter ‘Conclusions’.

Response: The chapter of Conclusions was added in line 273 in the revised manuscript.

-line 273:

5 Conclusions

The volatiles in the peels of 66 citrus germplasms respecting four citrus species were used for biomarker mining, and 30 potential biomarkers with different accumulation patterns in different citrus species were chosen by using PLS-DA. The β-elemene, valencene, nootkatone and limettin with the highest VIP values were chosen as biomarkers for the identification. It attained the accuracy of 90.0%, 96.7%, 96.7%, and 100% in loose-skin mandarin, sweet orange, pomelo and lemon, respectively. These biomarker compounds maybe responsible for the specific aroma in different citrus species. This method was a novel and effective alternative approach to identifying citrus genetic resources by biomarkers.

Reviewer 2 Report

Referee report for the manuscript Molecules #660556_v1:
„Volatile Compounds in Fruit Peels as Novel Biomarkers for the Identification of Four Citrus Species”

by Haipeng Zhang, Huan Wen, Jiajing Chen, Zhaoxin Peng, Meiyan Shi, Mengjun Chen, Ziyu Yuan, Yuan Liu,Hongyan Zhang and Juan Xu

The manuscript describes the use of volatile compounds in fruit peels as novel biomarkers for the discrimination and/or identification of four citrus species. It is carefully prepared and presents the main findings of this research in a clear and understandable way.

As far as this can be judged from the “Materials and Methods” section, the experimental procedure is sound. It appears that the present study is actually based on a study which was experimentally performed, evaluated and published by the same laboratory in 2017 (see reference [8]). So, apparently no new experiments have been performed, but instead the experimental data has been subjected to a data analysis in greater depth.

A smaller part of the results has thus already been discussed in the preceding publication and led to the identification of 34 distinctive volatiles which allowed the classification of different citrus germplasts via HCA (hierarchical clustering analysis) which agreed well with classic citrus taxonomy.

The current manuscript extends this classification by identifying suitable volatile biomarkers that allow the largely correct classification of the four citrus germplasts sweet orange, loose skin mandarins, pomelo and lemon.

Given that this may represent an easily applicable and efficient method for the discrimination of citrus germplasts, it is suggested to publish the manuscript with no alteration to its scientific contents.

Only few technical issues should be considered before the manuscript can finally be accepted:

The precision given in the abstract and in the manuscript for the accuracy of classification is reported in an unrealistically high way, e.g. 96.67% (round to max. 1 digit after the decimal point). Abstract and later: I think that the term “pummelo” is not idiomatic: “pomelo” is more frequently used in English language. L.35: rephrase (plural): “between plants with their…” L.53: rephrase “is quite similar but very different” -> what does this mean ? L.73: rephrase: “large scale volatile data” -> “large scale volatile compound data” L.80: A Venn plot is perhaps not generally known. Explain ! Experimental part: The correct classification of the citrus germplasts depends strongly on the correct assignment of the individual volatiles, and in particular, the terpenic compounds. As the mass spectra of particularly the terpenes are extremely similar; I was wondering how the correct assignment of the individual peaks was done and checked. Did the authors run chromatograms with authentic standards ? Did they use retention indices ? L.203: The abbreviations “SSR” and “SNP” are not generally known and shall be explained. The authors are also asked to comment on whether they have observed any temporal changes of the volatiles: Do the volatile compound patterns change when the samples are stored for a shorter or longer period of time, and is the classification affected by this ? The format of the references is not completely consistent. Please check and make uniform !

Author Response

Point1: The precision given in the abstract and in the manuscript for the accuracy of classification is reported in an unrealistically high way, e.g. 96.67% (round to max. 1 digit after the decimal point).

Response: Thanks for your suggestions. The accuracy of classification was corrected in the revised manuscript.

Point2: Abstract and later: I think that the term “pummelo” is not idiomatic: “pomelo” is more frequently used in English language.

Response: The “pummelo” was corrected as “pomelo” in the revised manuscript and supplemental materials.

Point3: L35: rephrase (plural): “between plants with their…”

Response: The “plant” was corrected as “plants” in the revised manuscript.

-line 35: volatile compounds play important roles in the interactions between plants with their environment [9] via attracting the pollinators, defending pathogens and herbivores [10-12].

Point4: L53: rephrase “is quite similar but very different” -> what does this mean ?

Response: the aroma of fruit are quite similar in different sweet orange germplasms, but different from loose-skin mandarin, pomelo and lemon accessions. The sentence was revised in the new manuscript.

-line 53: For example, the aroma of fruit are quite similar in different sweet orange germplasms, but different from loose-skin mandarin, pomelo and lemon accessions.

Point5: L73: rephrase: “large scale volatile data” -> “large scale volatile compound data”

Response: The “large scale volatile data” was corrected as “large scale volatile compound data” in the revised manuscript.

-line 73: In the study, large-scaled volatile compound data from 66 citrus germplasms representing 4 citrus species were used to mine the potential species-specific volatile biomarkers.

Point6: L80: A Venn plot is perhaps not generally known. Explain !  

Response: The sentence in L.80 was corrected in the revised manuscript.

-line 81: Totally 89 volatile compounds (see Supplementary Materials) from four citrus species were used to analyze, the Venn plot indicated that all of them were detected at least in two citrus species, and 59 volatile compounds were commonly detected in four species.

Point7: As the mass spectra of particularly the terpenes are extremely similar; I was wondering how the correct assignment of the individual peaks was done and checked. Did the authors run chromatograms with authentic standards ? Did they use retention indices ?

Response: The raw data were downloaded from our previous studies, 89 volatile compounds were used in this study, among of them, 40 were identified based on authentic standards. Retention indices were calculated using a homologous series of n-alkanes (C7-C30).

We agree with the reviewer’s comment, the mass spectra of terpenes are extremely similar, but some of volatile compounds were hardly to obtain the authentic standards, only were identified based on NIST Mass Spectral Library and retention indices. In order to ensure the accuracy of identification, all of those compounds were identified manually, for each compound; the MS matches scored at least 700 by spectral similarity, and selected the one with the highest probability.

Point8: L203: The abbreviations “SSR” and “SNP” are not generally known and shall be explained.

Response: The “SSR” and “SNP” were corrected in the revised manuscript.

-line 203-206: So, the use of volatile compounds as biomarkers for the identification of citrus germplasms were reliable, and the method was cheaper, simpler, faster and easier, might serve as a alternative or primary method before the molecular methods (Simple Sequence Repeats and Single-Nucleotide Polymorphisms).

Point9: The authors are also asked to comment on whether they have observed any temporal changes of the volatiles: Do the volatile compound patterns change when the samples are stored for a shorter or longer period of time, and is the classification affected by this?

Response: The contents of volatile compounds will have a little change during storage, more obvious changes will happen when stored for a longer period of time, but the characteristic or specific aroma is retaining. So, it does not affect the classification results.

Point10: The format of the references is not completely consistent. Please check and make uniform !

Response: The format of the references was corrected according to the reference formatting guide in the revised manuscript.

Round 2

Reviewer 1 Report

  The article can be published in its current form.